# Heterogeneous, delayed-onset killing by multiple-hitting T cells: Stochastic simulations to assess methods for analysis of imaging data

**Richard J. Beck**[ID]**, Dario I. Bijker**[ID]**, Joost B. Beltman**[ID] *

Division of Drug Discovery and Safety, Leiden Academic Centre for Drug Research, Leiden University, Leiden, The Netherlands

* j.b.beltman@lacdr.leidenuniv.nl

**Data Availability Statement:** All relevant data are within the manuscript and its Supporting Information files.

## Abstract

Although quantitative insights into the killing behaviour of Cytotoxic T Lymphocytes (CTLs) are necessary for the rational design of immune-based therapies, CTL killing function remains insufficiently characterised. One established model of CTL killing treats CTL cytotoxicity as a Poisson process, based on the assumption that CTLs serially kill antigen-presenting target cells via delivery of lethal hits, each lethal hit corresponding to a single injection of cytotoxic proteins into the target cell cytoplasm. Contradicting this model, a recent *in vitro* study of individual CTLs killing targets over a 12-hour period found significantly greater heterogeneity in CTL killing performance than predicted by Poisson-based killing. The observed killing process was dynamic and varied between CTLs, with the best performing CTLs exhibiting a marked increase in killing during the final hours of the experiments, along with a "burst killing" kinetic. Despite a search for potential differences between CTLs, no mechanistic explanation for the heterogeneous killing kinetics was found. Here we have used stochastic simulations to assess whether target cells might require multiple hits from CTLs before undergoing apoptosis, in order to verify whether multiple-hitting could explain the late onset, burst killing dynamics observed *in vitro*. We found that multiple-hitting from CTLs was entirely consistent with the observed killing kinetics. Moreover, the number of available targets and the spatiotemporal kinetics of CTL:target interactions influenced the realised CTL killing rate. We subsequently used realistic, spatial simulations to assess methods for estimating the hitting rate and the number of hits required for target death, to be applied to microscopy data of individual CTLs killing targets. We found that measuring the cumulative duration of individual contacts that targets have with CTLs would substantially improve accuracy when estimating the killing kinetics of CTLs.

## Author summary

The immune system plays an important role in controlling infections and tumours. Knowledge about the mechanisms through which the immune system accomplishes this

**Funding:** This work is supported by a Vidi grant from the Netherlands Organisation for Scientific Research (NWO; www.nwo.nl/en; grant 864.12.013 to J.B.B.). Funders played no role in the design or execution of this study.

**Competing interests:** The authors have declared that no competing interests exist.

can be exploited to develop immunotherapies. A pivotal mechanism involves killing of target cells by Cytotoxic T Lymphocytes (CTLs), yet limited quantitative knowledge on this process has so far been obtained, especially with respect to the killing capacity of individual CTLs. Recent results suggest that single CTLs exhibit substantial heterogeneity in killing capacity, even amongst clonal CTL populations. Here, we developed stochastic simulations of single CTLs killing small populations of target cells, showing that multiple-hitting can indeed lead to apparently heterogeneous killing kinetics between otherwise identical CTLs. We subsequently generated realistic artificial data using spatial simulations to study how multiple-hit killing parameters could be retrieved from future *in vitro* or *in vivo* time-lapse imaging data. Killing parameters were not identifiable when only data on number of killed target cells over time was available. Instead, we show that extraction of killing parameters is substantially improved if the cumulative contact times of CTLs with both killed and surviving target cells are monitored over time, and we offer an approach to fit such data in the future.

## Introduction

Cytotoxic T Lymphocytes (CTLs) are key effectors in the adaptive immune response, therefore CTL function—or lack thereof—is relevant in many pathologies. A greater quantitative understanding of CTL effector function will aid in interpretation of prior experiments and should yield useful insights for the treatment of diseases in the future. However, the rate at which CTLs kill infected or malignant cells remains poorly characterised. Estimates of CTL killing based on *in vitro* and *in vivo* CTL killing assays vary, with some variation explained by e.g. different susceptibility of target cells to CTL killing or the type of antigen expressed by the targets [1,2]. Moreover, especially *in vivo* the presence of stimulatory or suppressive factors and difficulty in controlling or estimating the ratio of CTLs to target cells at the site of killing might confound CTL killing estimates [1,2].

As a frequently discussed example, consider the *in vivo* CTL killing assay of Barber *et. al.* [3], in which CTLs demonstrated rapid killing against Lymphocytic Choriomeningitic Virus (LCMV). Although Barber *et. al.* initially estimated that CTLs took 15 minutes to kill targets, subsequent modelling studies based on the same data have estimated much faster killing rates [4,5,6], with one study implying an expected target survival time of 16 seconds after contact from a CTL [4] (see also [1] for a detailed summary of these estimates). Given that killing in those experiments was perforin-dependent[3], these fast estimates seem to contradict recent *in-vivo* imaging showing that the perforin-dependent killing process requires a minimal contact time. For example, long-lasting (median: 80s) calcium fluxes linked with CTL killing of virally infected cells occurred, on average, 480s (median) after CTLs established contact with virally infected targets[7]. Such killing times of around 10 minutes are consistent with the duration of killing events that can be observed in various supplemental videos elsewhere [8,9]. Given this lower bound it is difficult to see how solely granule-mediated killing could plausibly lead to killing rates in excess of ~6 hour$^{-1}$, even in optimal situations where CTLs are not limited in their supply of targets and do not require time to search for new targets between killing events.

A major limitation of many prior estimates of CTL killing is that analysis is performed on population level data in *in vivo* settings, with no direct measurements of the killing process. This approach has a number of drawbacks: First, it can be challenging to accurately assess the frequency of CTLs and target cells. Second, other immune cells may contribute to the killing

process, confounding estimates of the true CTL killing rate. Third, the processes underlying CTL killing are complex and it may be insufficient to describe them with a single, time invariant rate constant. Indeed, recent observations have indicated that target cells may require multiple hits before death either *in vitro* [10], or *in vivo* [7]. We have previously shown that such multiple-hitting can lead to a time-increasing killing kinetic when CTLs are exposed to fresh targets [11,12], further complicating the killing rate estimation procedure.

Besides analysing CTL killing performance at the population level, a potentially useful approach is to analyse CTL killing at the single cell level. Such analysis can yield greater insights into the dynamics of the killing process. This was exemplified in studies undertaken in the 1970's in which the killing kinetics of CTLs conjugated with 1–4 EL4 tumour cell targets were examined under the microscope for a period of 3 hours [13,14]. Subsequent mathematical analysis of these studies indicated that the CTL killing process was well described as a Poisson process [15], indicating that CTLs kill targets sequentially rather than simultaneously. This analysis allowed the authors to conclude that CTL killing was mediated by secretory lysosomes, several years before this was demonstrated conclusively [16]. The aforementioned studies also revealed that the rate of CTL killing was not diminished after target lysis, an observation which led the authors to deduce that CTLs were able to discriminate between viable and killed targets. More recently, *in vitro* studies of individual natural killer cells have shown that killing occurs via both granzyme and death receptor mediated pathways, each having different kinetics [17,18].

Despite the utility of studying CTL killing at the single-cell level, there remains a shortage of *in vitro* CTL killing studies with statistical power sufficient to check the validity of the Poisson model first proposed in the 1980's by Perelson et. al. [15]. Recently one such a study was performed: Over a 12 hour period, image-based killing measurements were taken from human-derived CTL clones, each CTL being separately confined within small micro-wells that contained an excess of JY target cells [19]. During the studied time period, the killing rate of CTLs was dynamic, exhibiting a marked increase in the final hours of the experiment. The total number of targets killed per CTL was overdispersed compared to the Poisson distribution, implying greater heterogeneity between individual CTL killing performance than anticipated. Vasconcelos et. al. (Vasconcelos et al. 2015) found the data was well described by a Poisson mixture model, and they postulated the existence of a subset of "high rate killers" comprising 30% of the population that emerged 8–10 hours after first exposure to target cells. However, no mechanistic explanation could be found to explain this result, despite a search for membrane markers that might identify and/or explain the variability of CTL killing characteristics.

We hypothesised that a requirement for "multiple hits" to kill targets before apoptosis induction might explain heterogeneous killing amongst clonal CTLs *in vitro*. Perelson et. al. [20] previously considered the possibility of multiple-hitting, noting however that such a model was excessively complex to describe the limited experimental data available at that time. Recent evidence has directly shown that multiple-hitting does occur at least in some settings [7,10,21], and our previous modelling work has demonstrated that multiple-hitting can indeed lead to population-level killing kinetics increasing over time when CTLs are exposed to fresh targets [11]. Therefore, we here used stochastic simulations to investigate the compatibility of the multiple-hitting hypothesis with the findings of Vasconcelos et. al. [19]. We found that multiple-hitting was indeed able to explain the late onset, high-rate bursting kinetic of individual CTLs, with physiologically plausible parameters. We also highlight that multiple-hitting is expected to lead to a complex dependence of realised killing rate upon the number of available targets and on the ability of individual CTLs to form and abort conjugates with target cells. We subsequently developed spatially explicit, agent based simulations of CTLs killing targets in

micro-wells as a means of generating realistic yet noisy artificial data and assessing methods of recovering CTL hitting parameters from future microscopy data. Using these spatial simulations, we demonstrate how parameter estimation is substantially improved if contacts of individual targets with CTLs can be tracked throughout the duration of the experiments.

## Results

### Multiple-hitting CTLs exhibit heterogeneous late onset killing

We first sought to establish whether the multiple-hitting hypothesis was a feasible explanation for the heterogeneous, delayed onset, "burst" killing kinetics observed and defined by Vasconcelos *et. al.* [19]. In brief, these high rate killer CTLs were a subset among a clonal population whose killing suddenly accelerated after 8–10 hours of experimentation, with no explanation readily apparent (Methods). In the current study, we used Monte Carlo simulations of individual CTLs killing targets to identify conditions under which multiple-hitting might lead to heterogeneous, "burst" killing. In these Monte Carlo simulations, CTLs hit targets at a constant rate $\lambda$, then targets died after receiving $\eta$ hits. We simulated single- and multiple-hitting scenarios on the basis that the expected (mean) time for one target in contact with a CTL to be killed was 1 hour, i.e., we set $\lambda/\eta = 1$ (valid for entire Fig 1).

Firstly we simulated CTLs with $\eta = 1, 2$, or 10, with each simulation containing one CTL interacting with a single target. For such a strictly 1:1 CTL:target ratio, the waiting times for target death were gamma distributed with rate parameter $\lambda$ and shape parameter $\eta$ (Fig 1A). The gamma distributions (Fig 1B, top panel), together with their accompanying survival probability functions (Fig 1B, middle panel), define the hazard function (Fig 1B, bottom panel), which is the momentary rate of death experienced by a target, given that the target has already survived an interaction for some time, $t$. When $\eta = 1$, the hazard experienced by contacted targets does not change with time. In contrast, when $\eta > 1$ the hazard experienced by contacted targets increases over time, as contacted targets become increasingly likely to have received ($\eta - 1$) hits and thus be killed by the next hit. For the case where CTLs interact with targets in a strictly 1:1 ratio, the gamma($\eta, \lambda$) distribution parameters could be estimated from the mean and variance of the samples of the waiting time ($y_o$): $\bar{y_o} = \frac{\eta}{\lambda}$ and $Var(y_o) = \frac{\eta}{\lambda^2}$ (Fig 1C).

We next extended our Monte Carlo simulations to allow CTL:target interactions in a 1:n ratio, for variable numbers of targets, $n$. CTLs were individually assigned their initial number of targets by drawing $n$ from a Poisson distribution, with mean $\bar{n} = 16$ (Fig 1D, blue bars). The total number of targets killed by a CTL during one simulation, $x$, should also follow the Poisson distribution, if the killing rate of each simulated CTL would be the same. Moreover, the mean and variance should be approximately equal for any set of Poisson distributed samples. Therefore, observation of a ratio $\frac{var(x)}{\bar{x}} > 1$ for a set of killed targets would imply that the killing was more heterogeneous than expected under Poisson assumptions. For single-hit killing ($\eta = 1$), the variance of the 12 hour killing samples was in fact slightly below the mean (Fig 1D, upper row), resulting from some simulations where CTLs killed all their targets before the simulation had finished. However, for $\eta = 2$ (Fig 1D, central row), the variance approached the mean and for $\eta = 10$ far exceeded the mean (Fig 1D, bottom row). In the latter case, a bimodal distribution occurred, which could be interpreted as a subpopulation of high-rate killers, yet importantly such a population did not exist in our simulations.

In our simulations, the additional variability in killing performance of multiple-hitting CTLs was due to the allocation of subsequent hits amongst several different targets. When a group of targets share hits evenly, the time for a specified target to be hit is proportional to the number of other targets sharing. This has no effect on the killing rate observed if $\eta = 1$, so the mean killing rate for our simulated single-hitting CTLs initially remained constant over time

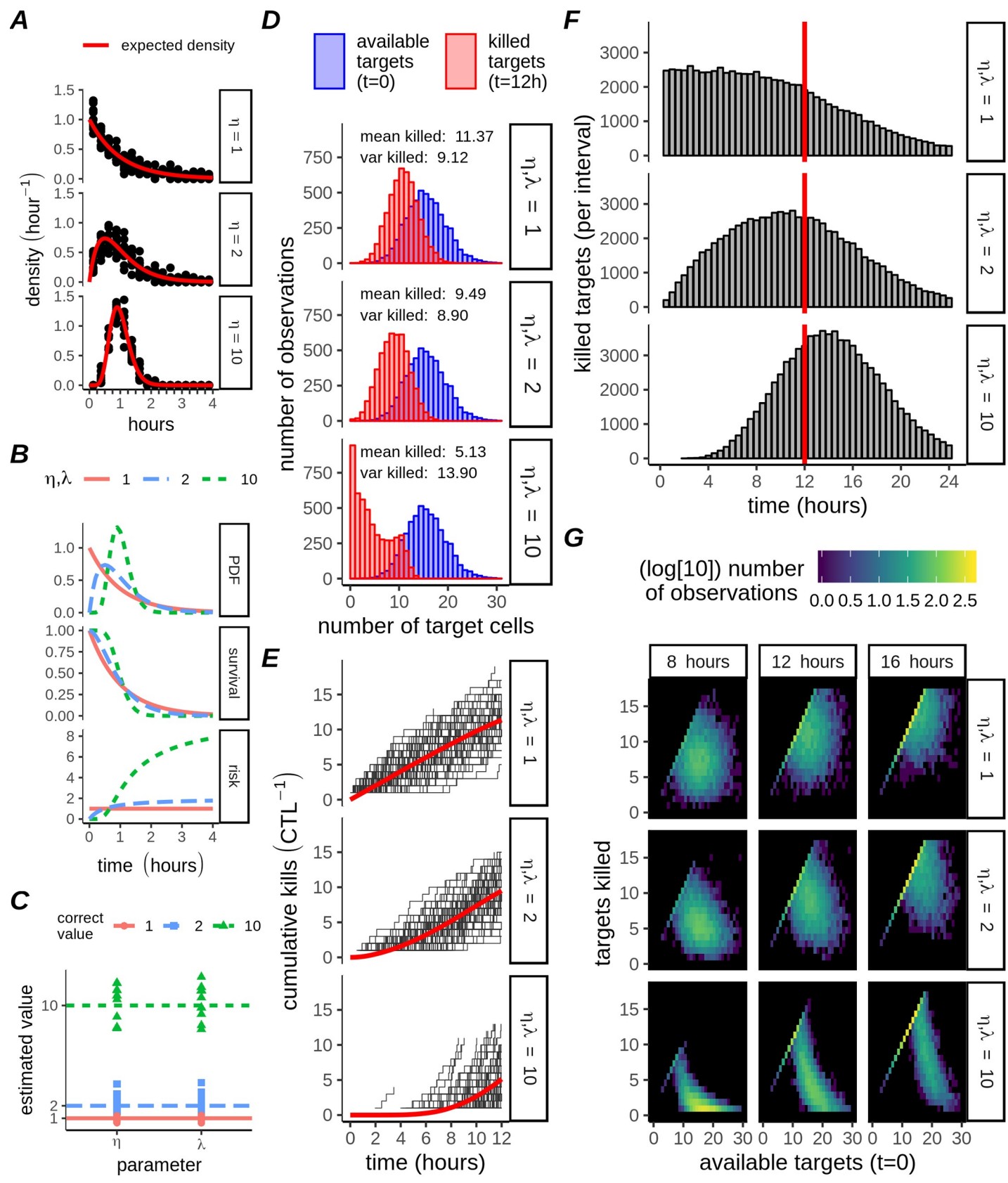

**Fig 1. Killing kinetics and heterogeneity of multiple-hitting CTLs.** A) Gamma probability density functions describing expected time for a CTL to kill 1 target in monogamous contact (red lines). Each point represents the sample killing density of one series of simulations ($N_S = 10$), each series comprising $N_w = 100$ CTL:target pairs. Observations were binned at 15 minute intervals. B) Theoretical Gamma probability density function (PDF), survival function, and hazard function for different values of $\eta$ as indicated. C) Estimation of parameters from simulations in panel A, by equating the first two moments (the mean and variance) with their estimators. D) Distribution of targets killed per CTL at 12 hours (red bars), with Poisson distributions for the initial number of targets (using the same initialising distribution for all $\eta$; blue bars). Each panel contains results from $N_w = 5000$ CTLs for different $\eta$, as indicated by facet labels on the right. Text inside panels indicates the mean and variance of the killed targets. E) Cumulative killing performance of $N_w = 100$ members (thin black lines) of the population shown in D; the red line is the mean calculated for the entire population ($N_w = 5000$). F) Distribution of target killing times over extended (24 hours) simulations with CTL parameters matching D, with the 12 hour censorship indicated by a red line ($N_w = 5000$, bars are kills per 30 min interval). G) Heatmap of the probability density for each simulation in C-D. Observations were binned according to unique combinations of the initial number of targets (individual columns), together with the number of killed targets at the indicated interval (individual rows). Thus, summing across columns will recover the initial Poisson distribution (blue bars in D), and summing across rows will produce the distribution of killed cells at the indicated time (e.g red bars in D at 12 hours).

(Fig 1E and 1F, top row; time<8h), gradually decreasing as some CTLs eliminated all their targets (Fig 1E and 1F, top row; time>8h). In contrast, hit sharing in the case of multiple-hitting CTLs led to a delayed onset of killing (Fig 1E and 1F), with the length of the delay dependent on the number of targets sharing hits (compare Fig 1A and 1B with single targets to Fig 1F with multiple targets, for identical $\eta$). The interaction between $\eta$ and the number of initial targets can also be understood from heatmaps of targets killed (Fig 1G). The expected cumulative number of kills increases over time for $\eta, \lambda = 1$, but this increase is independent of the initial number of targets except for the censorship implying a maximum target number that can be killed. For $\eta, \lambda = 10$, the dependency of the observed kills on the initial number of targets is very clear, with killing happening earlier in those wells with initially fewer targets. Moreover, these effects did not only depend on the initial number of targets. When we performed simulations with the hitting rate $\lambda$ a random variable, this in turn increased the variability of killing amongst multiple-hitting CTLs to a greater extent than was the case for single-hitting CTLs (S1 Fig), implying that the killing of multiple-hitting CTLs is more sensitive to environmental variables than the single-hitting CTLs. Taken together, these results imply that multiple-hitting CTLs could explain both heterogeneous and delayed onset killing among clonal CTL populations.

## Multiple-hitting is not identifiable based on population killing statistics only

We asked if population-level killing statistics (as e.g. examined in [19])) could be used to identify the hitting parameters ($\lambda$ and $\eta$) of CTLs. In our previous simulations (Fig 1) we studied a scenario of simultaneous risk for target cells, yet this may be an oversimplification. For example, due to physical constraints the number of targets CTLs can simultaneously contact and thus hit must be limited. Therefore, we extended our *1:n* Monte Carlo simulations to allow dynamic contacts between CTLs and targets, in order to check how the parameter estimates ($\lambda$ and $\eta$) would be impacted. To achieve this, we included an additional state for target cells, now distinguishing between targets that are contacting the CTL, versus those not in-contact (Fig 2A; Methods). The killing kinetics realized by CTLs in these dynamic simulations indeed differed from those in our previous simulations (Fig 1), where all the targets shared risk and so the killing rate of each CTL was dictated by the total number of yet-living targets. In contrast, for our Monte Carlo simulations allowing dynamic conjugate formation, only the targets presently being contacted were relevant. Here, small bursting events occurred throughout the simulations, which was the result of accumulating hits followed by rapid sequential killing among a subset of contacted targets (Fig 2B).

Using our simulations including dynamic conjugate formation we searched for parameters consistent with the statistics reported previously [19] concerning high rate "burst killing"

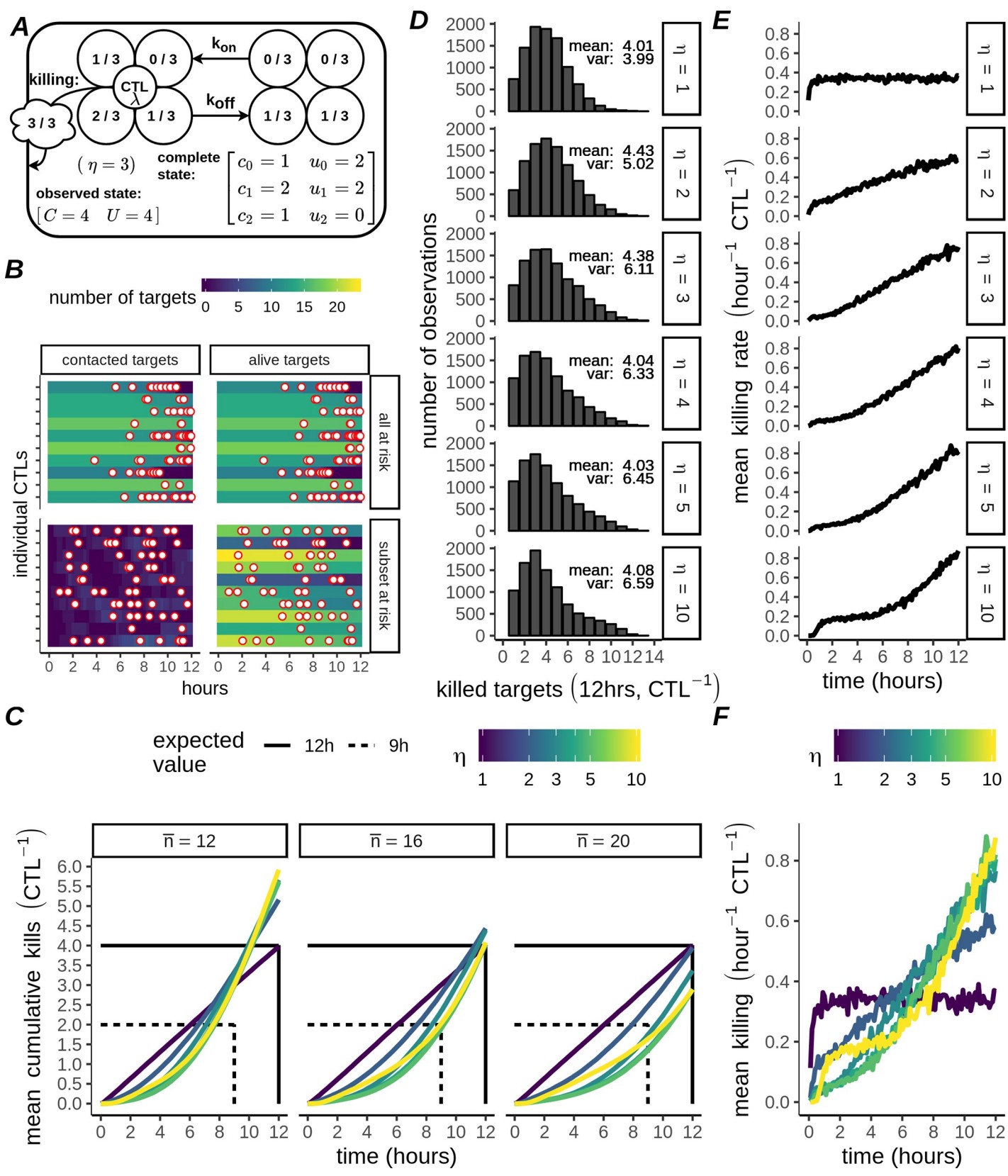

**Fig 2. Burst killing and non-identifiability of dynamically interacting, multiple hitting CTLs.** A) Schematic of the dynamic model (example with $\eta = 3$). Target cells are represented by circles containing fractions (numerator: hits received; denominator: $\eta$). The observable state $[C\ U]$ consists of the total number of contacting, $C$, and non-contacting targets, $U$. The complete state of the system is represented by a matrix, with $\eta$ rows indicating the number of hits received (subscript) and with columns indicating whether the target is contacting ($c_i$) or non-contacting ($u_i$). B) Measured killing events (red dots, filled white) during Monte Carlo simulations with $N_w = 10$ CTLs and the number of targets drawn from a Poisson distribution with mean $\bar{n} = 16$ (top panels; $\lambda, \eta = 10, k_{off} = 0, k_{on} \to \infty$), or a subset of targets at risk (bottom panels; $\lambda, \eta = 10, k_{on} = 1 hr^{-1}, k_{off} = 0.3 hr^{-1}$). Each horizontal strip is one single simulation, the right panel strips are colored according to the total number of alive targets and the left panel strips are colored according to the number of targets that are in contact with a CTL. C) Each line (coloured according to $\eta$) is the mean cumulative killing over time ($CTL^{-1}$) from $N_w = 10^4$ CTLs, simulated using parameters estimated by fitting the case with $\bar{n} = 16$ targets (central column). Straight lines show target values for fitting. All parameters except $\bar{n}$ are constant across columns. D-F) Measured statistics within simulations with $\bar{n} = 16$ shown in the central column of C. Shown are the distribution of killed targets after 12 hours, with the mean and variance as indicated for each $\eta$ (D), and the mean killing rate over time for CTLs grouped by $\eta$ and shown either separately (E, rows), or together (F, colors), calculated as $(kills \cdot (6\ min \cdot N_w)^{-1})$.

CTLs. For fitting we used the reported group mean (4) and variance (6.9) of the number of killed targets per CTL over 12 hours. Additionally, we aimed for a breakpoint in the mean killing rate such that half (2) of the observed kills occurred in the interval 0–9 hours and the other half in the interval 9–12 hours (see Methods). We performed this fit using different values for $\eta$ (ranging from 1–5, or 10), and a Poisson variable with $\bar{n} = 16$ for the initial number of targets (S2 Fig). We obtained good fits for different values of $\eta$; in particular for all $\eta > 2$ the cumulative killing was very closely matched (Fig 2C, intersecting lines for $\bar{n} = 16$). Moreover, for values $\eta > 3$ the fits to the mean and variance for cumulative targets killed at 12 hours were all similarly close to their target values of 4 and 6.9, respectively (Fig 2D). Some differences for different $\eta$ were apparent, for example as the number of hits increased towards $\eta = 10$ the breakpoint marking transition from low to high rate was more distinct (Fig 2E). However, overall differences between $\eta$ were quite small (Fig 2F), and many simulated CTLs were required for these differences to emerge consistently (at least $N_w = 10^3$ CTLs). Our results were also sensitive to the distribution for the initial number of targets per CTL: for simulations with $\bar{n} = 12$ or 20, substantial differences in the cumulative kills over time occurred (Fig 2C). Thus, we conclude that multiple-hitting is not only qualitatively, but also quantitatively consistent with the experimental results reported previously (Vasconcelos et al. 2015). However, our analysis shows that the mean and variance of the killing process measured for a group of CTLs are insufficient statistics to determine the number of hits CTLs require to kill targets, so CTL:target interactions should be explicitly accounted for if killing due to multiple-hitting is to be modelled accurately.

## An Agent Based Model of Multiple-hitting CTLs to test methods for estimation of killing parameters

Since we found that in many situations the true hitting parameters for CTLs could not be determined based on group level killing mean and variance, we sought methods to compare the likelihood of different hitting models. We did not wish to consider a particular model for the process of CTLs finding targets, preferring a method that could be applied to determine the CTL hitting behaviour in general situations (i.e., in the absence of knowledge on contact dynamics). As a framework for testing we employed an agent based cellular Potts model (CPM) to generate 2D simulations of CTLs interacting with and killing targets (Fig 3A). The resulting datasets were visually similar to realistic microscopy data and could be used to investigate methods for recovering the hitting parameters ($\eta$ and $\lambda$) of CTLs from experimental data under various conditions. For all CPM simulations we maintained the same underlying gamma model of CTL hit generation as was used for our Monte-Carlo simulations (Figs 1 and 2), however we made several modifications that would lead to different (yet not predictable _a priori_) distributions of hits amongst targets. Specifically, instead of allocating hits to all contacted targets with equal probability, target risk of receiving a hit was proportional to the

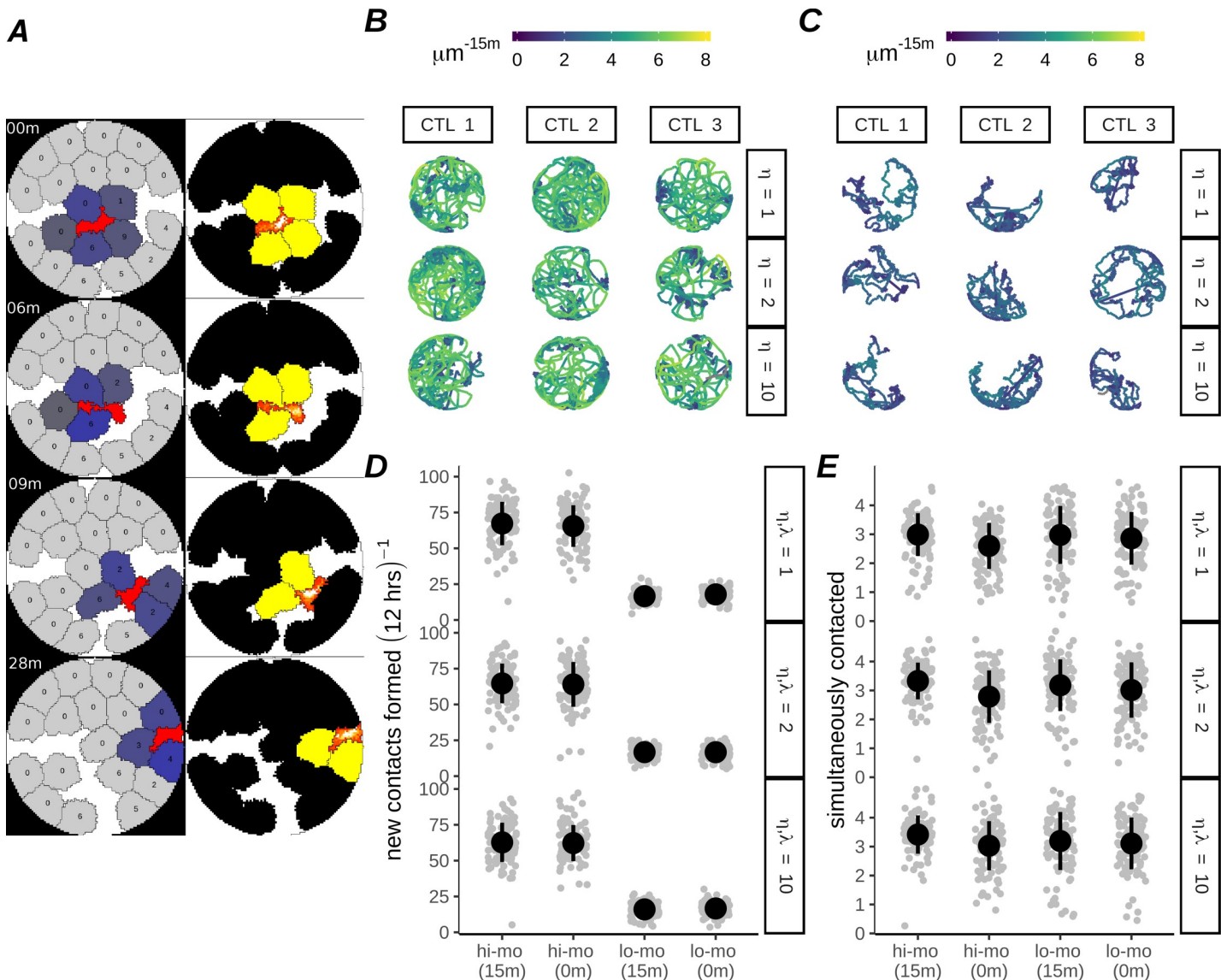

**Fig 3. Characterisation of high- and low-motility *in silico* CTLs within CPM simulations.** A) Still images of a high motility CTL with 15 minute minimal hitting time, interacting with targets. Left color scheme: CTLs are red, uncontacted targets are grey, and contacted targets have various shades of blue based on their share of total CTL:target interface, which determines their probability of receiving a hit. Targets are overlaid with the number of hits they have received. Right color scheme: Lattice sites inhabited by the CTL are colored according to actin activity [22]. Targets are black, turning yellow after 15 minutes of continuous contact with the CTL. Elapsed simulation time is displayed in the upper left corner of the stills, presented in minutes since the first frame shown. B-C) Track plots showing movement of 3 randomly sampled CTLs of high (B) and low (C) motility throughout a simulation, for simulated η as shown. D-E) Frequency at which CTLs form new conjugates (D) and mean number of simultaneously contacted targets per CTL (E) for low- and high-motility CTLs. Plots are based on 100 simulations per condition, with each dot representing one CTL, and circles and error bars indicating mean +/- SD.

length of the interface between CTL and target at the moment of hit generation (Fig 3A, target coloring on left images indicates interface length). We also considered the effect of a lower bound on the time required for a CTL to complete a hit, by introducing a delay condition that prohibited targets from being hit within an initial time window after contacting a CTL, which was reset every time the target broke contact with the CTL (Fig 3A, target coloring on right images). Note that the delay condition was applied per target and therefore does not preclude the possibility of CTLs hitting other contacted targets simultaneously.

Finally we varied CTL migration to create two groups of CTLs which we termed "high-motility" (Fig 3B, S1 Video) or "low-motility" (Fig 3C, S2 Video) CTLs. For both motility conditions the migration of the CTLs was influenced by the presence of the targets, as CTLs became corralled by surrounding targets. The difference between these models was that high-motility CTLs exhibited an increased propensity to break free from confinement and roam the well. This roaming ensured that over the course of 12 hours the high-motility CTL made new contacts with far greater frequency than low-motility CTLs (Fig 3D), although the average number of simultaneously contacted targets at any time was similar (Fig 3E). Thus, the high-motility CTL is expected to approach the previously modeled 'all targets at risk' scenario more closely than the low-motility CTLs.

We used the CPM model to simulate CTLs (with $\eta, \lambda = 1, 2,$ *or* 10), in either high- or low-motility scenarios. The total amount of targets killed by each CTL depended on the interaction between the parameters $\lambda$ and $\eta$, the CTL motility, and the presence or absence of the delay condition. In particular, the combination of high motility plus 15 minute delay resulted in a substantial decrease in killing in comparison to the other simulation groups, for all values of $\eta$ (Fig 4A). Together with the high rate of contact formation in that group (Fig 3D), this is consistent with targets spending significant time in transient contacts with the CTL, too short to result in successful hit delivery. The killing rate of the low-motility CTLs was initially greater than of high-motility CTLs, in particular for large $\eta$ (Fig 4B), due to the more stable nature of the contacts leading to greater accumulation of hits among the contacted targets (Fig 4C). High-motility CTLs reduced this deficit over the course of the simulations due to an accumulation of latent hits among uncontacted targets (Fig 4D). These spatial simulations therefore illustrate how CTL:target contact dynamics can play a role in determining killing performance. Moreover, since in these models CTLs with the same killing parameters—but different motility parameters—generated different killing kinetics, they are useful to test how underlying killing parameters might be recovered from microscopy data that are similar to data emanating from our realistic simulations.

## Estimating CTL hitting parameters through analysis of contact time and target survival

Since we found that hitting parameters $\eta$ and $\lambda$ could not be recovered via analysis of population averages only, we employed a parametric survival analysis to study the hazard experienced by individual targets contacting CTLs. Our analysis considers the different hazard functions identified earlier (Fig 1B), which distinguish CTLs on the basis of their intrinsic hitting rate $\lambda$ and the number of hits required for killing targets, $\eta$. Specifically, we analyse the cumulative duration of CTL:target contact events from the perspective of the target cells (example in Fig 5A). To take into account shared hazard amongst a set of co-contacting targets, we recorded for each sampled frame the statistic $\theta = (c)^{-1}$ (per-target), representing the probability that each separate target out of the subset of $c$ targets co-contacting the CTL is presently being hit (Fig 5B). Note that targets not in contact with the CTL were assigned $\theta = 0$. Subsequently, we integrated the $\theta$ values over time to arrive at a set of 'adjusted' contact times, $\tau$, for each target (Fig 5C), which takes into account uncertainty with respect to hitting of multiple co-contacted targets (Fig 5C). This approach has the advantage that no explicit account needs to be taken of the CTL-target interaction dynamics. Moreover, estimation of cellular contact times occurs already frequently in time-lapse imaging data [7,23], hence is feasible.

Applying the concept of adjusted contact times, $\tau$, on all our CPM simulations, we established maximum likelihood estimates for the hitting parameters within the simulations (S1 Text; S3 Fig). This yielded excellent estimates for the parameters in simulations without delay

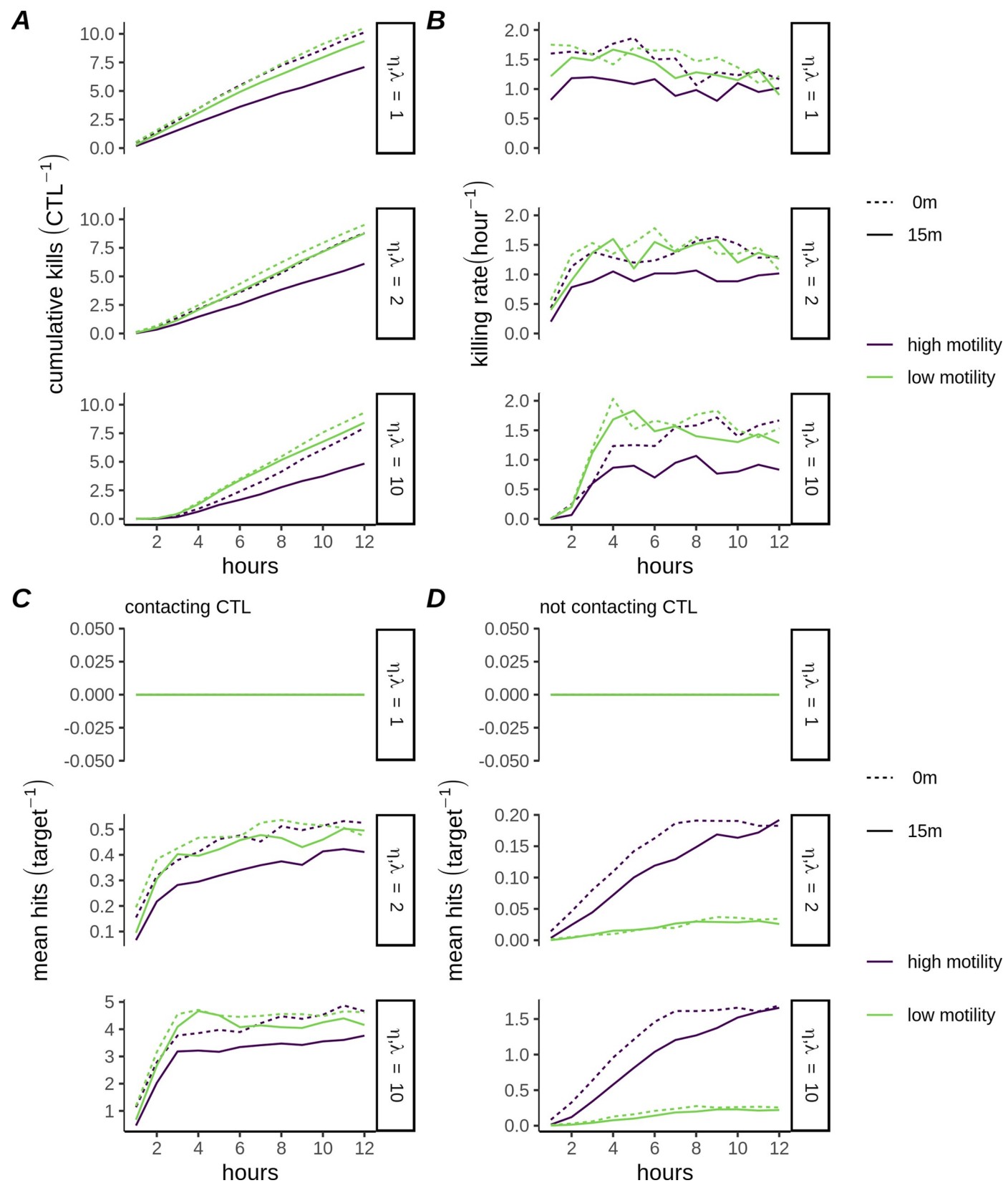

**Fig 4. Killing performance of multiple-hitting CTLs depends on motility.** A) Mean cumulative killing over time (CTL⁻¹) for CPM simulations of high- and low-motility CTLs ($\eta,\lambda$ = 1,2, *or* 10) B) Mean killing rate (CTL⁻¹) for each simulated condition in A. C-D) Mean number of hits received per target, sampled over targets currently contacting the CTL (C) or over targets not currently contacting the CTL (D).

(Fig 5D; $\eta_{CPM},\lambda_{CPM}$ indicate input CPM parameter values, and $\hat{\lambda},\ \hat{\eta}$ indicate estimated values). We also tested our method of parameter recovery by fitting our model to small subsets taken from the CPM simulations each containing only $N_w$ = 10 CTLs (Fig 5E), which led to good estimates. Additionally, we tested our model on sample data generated from a mixed dataset with two subpopulations of single-hitting CTLs, each with a different killing rate (S1 Text), in order to examine the high-rate-killer hypothesis put forth by Vasconcelos *et. al.* [19]. We found that the multiple-hitting model would not predict multiple-hitting unless multiple-hitting was indeed underlying the data, instead predicting a single-hitting population whose killing rate was the mean of the individual subpopulations (S1 Text; S4 Fig, S5 Fig). Thus, our maximum likelihood approach based on contact time monitoring can distinguish multiple-hitting from alternative hypotheses and is expected to work for a relatively small number of samples.

## Impact of a hitting threshold on killing parameter estimation

Given the time needed for formation of a cytotoxic synapse that is required for hit delivery, brief interactions between CTLs and targets may not contribute to killing. Taking such brief interactions into account in our parameter estimation may thus interfere with correct estimation. Therefore, we tested our parameter recovery on those CPM simulations wherein a 15 minute minimal bound (+15m) was set for the time CTLs required to successfully execute each hit upon a target. In these CPM simulations, we generally obtained robust estimates for the number of hits needed to kill targets, $\hat{\eta}$ (S6A Fig). However, after rounding to the nearest integer value for $\eta$, the estimated hitting rate parameter, $\hat{\lambda}$, was underestimated compared to the generating value ($\lambda_{CPM}$) in simulations with the 15m minimal hitting time. Since the realised killing was reduced in the 15m-delay simulations, particularly for high-motility CTLS (Fig 4), the estimated $\hat{\lambda}$ could be considered more appropriate than the generating value $\lambda_{CPM}$. Nevertheless, to investigate further we performed additional simulations, using high motility CTLs, with variable hitting delays in the interval between 0–15 mins (Figs 6A and S6B). We found that for the important boundary between single-hitting ($\eta$ = 1) or multiple-hitting ($\eta$ = 2), the estimated number of hits parameter $\hat{\eta}$ was accurately classified for limited delays of less than 15 minutes (Fig 6A, top row).

A particular advantage of a parametric survivorship analysis such as that we employ here is that, having estimated the hitting parameters ($\eta,\lambda$), we can revisit the sample data and ask whether different subsets of targets were killed according to our expectation. We selected the high-motility +15m simulations with $\eta_{CPM}$ = 1 for further study, since for this simulation group there was an ambiguous estimate of $\hat{\eta}$. For comparison, we also analysed the data from CPM simulations with multiple-hitting CTLs ($\eta_{CPM}$ = 2). First, we inspected the Kaplan-Meier estimates of the survival functions (Fig 6B, black lines) marking close agreement when the correct parameter estimate ($\hat{\eta}$ = 2) was applied to CPM data generated by multiple-hitting CTLs (Fig 6B bottom, blue line, $\eta_{CPM}$ = 2), but not when the incorrect $\hat{\eta}$ = 1 was applied (Fig 6B bottom, red line, $\eta_{CPM}$ = 2). For data generated by single-hitting CTLs, the Kaplan-Meier estimate lay exactly between the estimates using $\hat{\eta}$ = 1 or 2, yet the shape of the survival function over the entire length better matched that for the single-hitting estimate $\hat{\eta}$ = 1 (Fig 6B top, red line, $\eta_{CPM}$ = 1) than for the estimate $\hat{\eta}$ = 2 (Fig 6B top, blue line, $\eta_{CPM}$ = 1). Second, visual

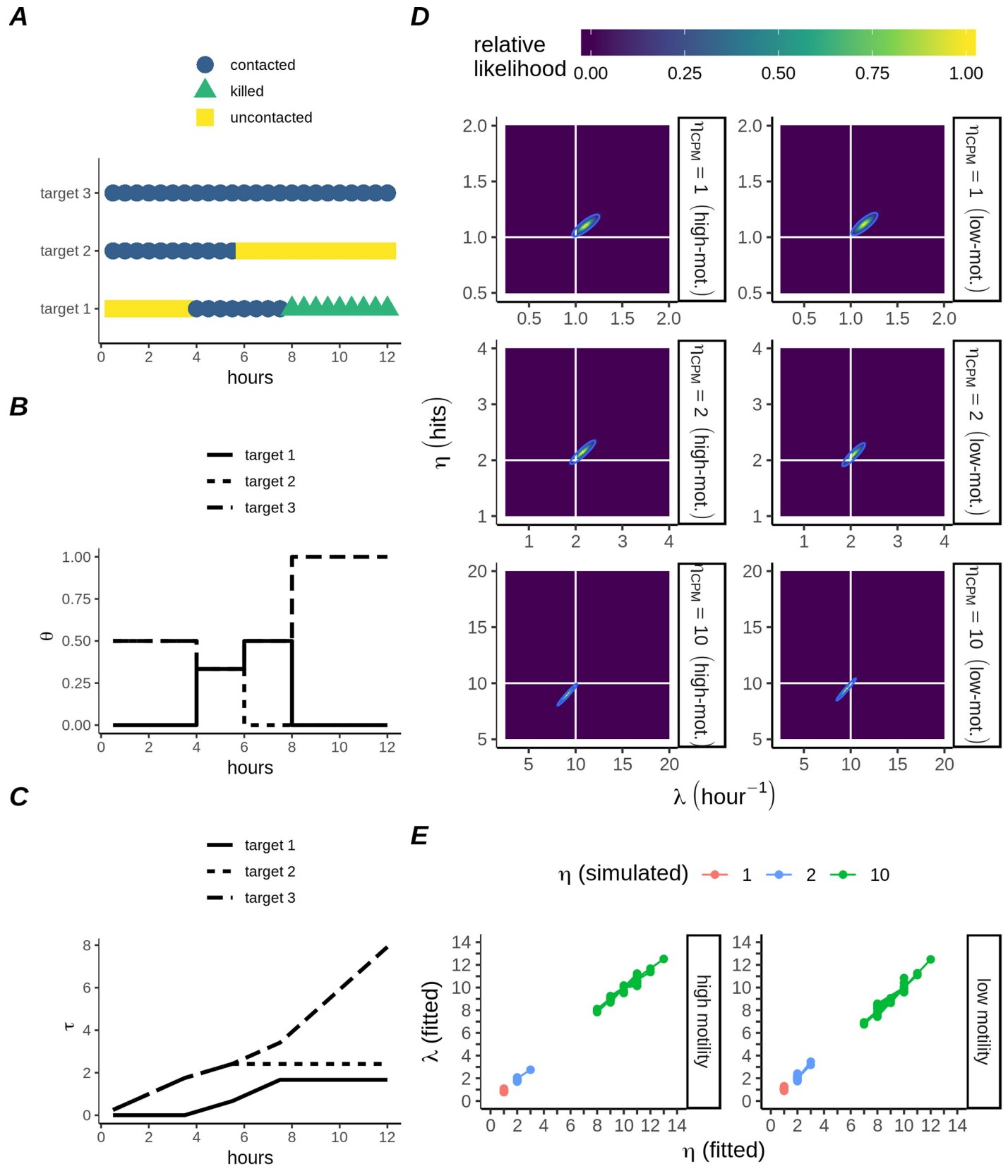

**Fig 5. Parameter retrieval for multiple-hitting CTLs based on adjusted contact time.** A) Hypothetical example illustrating sharing of subsequent CTL hits by target cells. Interaction history during a period of 12 hours for each of three target cells contacted by a single CTL, sampled at 30 minute intervals. B) Estimated probability (expressed as fraction $\theta$) that each target is being hit by the CTL, corresponding to the hypothetical interaction history shown in A. C) The quantity $\tau$ is defined as the cumulative sum over the course of the simulation of all sampled values of $\theta$ associated with each individual target. The samples resulting from interaction with this CTL include target 1, which was killed after a cumulative interaction period of ~1.7h, and targets 2 and 3, which remained alive after cumulative interaction periods of ~2.4h and ~7.9h, respectively. D) Heatmaps of the likelihood function around the maximum likelihood estimates for the killing parameters, in CPM simulations without hitting delay. Horizontal and vertical lines mark the values of the CPM parameters used to generate the data for each group. The boundary enclosing the 95% confidence region is also marked with a line. E) Results of fitting 30 randomly chosen subsets, each consisting of $N_w = 10$ simulations, of the CPM simulations without hitting delay.

inspections of the hazard experienced by individual targets throughout the simulations (Fig 6C), revealed that in many CPM simulations with $\eta_{CPM} = 1$ and a +15m hitting delay there was substantial killing of targets that had not yet undergone long interactions with the CTL (Fig 6C, first two grey bars), as would not be expected for multiple-hitting. Thus, both results (Fig 6B and 6C) supported $\hat{\eta} = 1$ as the most likely candidate for the data derived from simulations with $\eta_{CPM} = 1$. However, the most conclusive result was obtained by evaluating the mean hazard experienced by contacted targets according to either of the two candidate estimates for the number of hits ($\hat{\eta} = 1$ or 2). Integrating this value over the duration of the experiments (Fig 6D, black lines) led to predictions for the killing rate over time which closely followed the data whenever a correct estimate for $\hat{\eta}$ was applied (Fig 6D, comparing black and red lines in the upper panel, or black and blue lines below). In contrast, killing predictions from incorrect estimates of $\hat{\eta}$ were extremely poor, thus allowing for correct identification of the underlying $\eta$. Thus, our analysis shows that monitoring of cumulative interaction times between targets and single CTLs allows for proper estimates of the number of hits required for target cell death even when brief contacts between CTLs and targets cannot lead to hits, although the hitting rate may be underestimated in that case.

## Discussion

Here we have used stochastic simulations to show that 'multiple-hitting' is a plausible explanation for the heterogeneous and time-inhomogeneous killing activity recently observed for CTLs *in vitro* [19]. We showed that multiple-hitting leads to an increase in realised killing rate over time. Moreover, the extent of this late onset killing increases when more hits are required to kill targets, or when a greater number of antigen-presenting targets are simultaneously contacted. Furthermore, identical CTLs displayed varying killing performance depending on the number of targets available. Simulating CTLs with variable hitting rates, we also found that the killing performance of multiple-hitting CTLs is more heterogeneous than killing of single-hitting CTLs, given similar variation in underlying hitting rate. Overall, we conclude that multiple-hitting is sufficient to explain heterogeneous killing amongst clonal CTLs and there is no need to invoke an unobserved subpopulation of high-rate killers.

Given the dependence of the killing performance of multiple-hitting CTLs on several parameters that we describe here, we developed spatially explicit CPM simulations to assess methods for investigating whether multiple hitting occurs *in vitro* or *in vivo*. Our specific goal was retrieval of the hitting rate and number of hits required for CTLs to kill targets. Our model of dynamic conjugate formation can be conceptualised using Kendall's notation as an $M/E_r/1$ queue [24]. Within this framework there is 1 "server" (in our case the CTL), with markovian arrival times (M: in our case conjugate formation events), and Erlang distributed ($E_r$) "service times" which represent the killing process. It is known that for such models, the mean and variance (or any similar measure of variability) are insufficient for estimating the true parameters, and can only be used to approximate the distribution [24]. Instead of using population-level killing statistics, we were able to accurately recover model parameters from the CPM data by

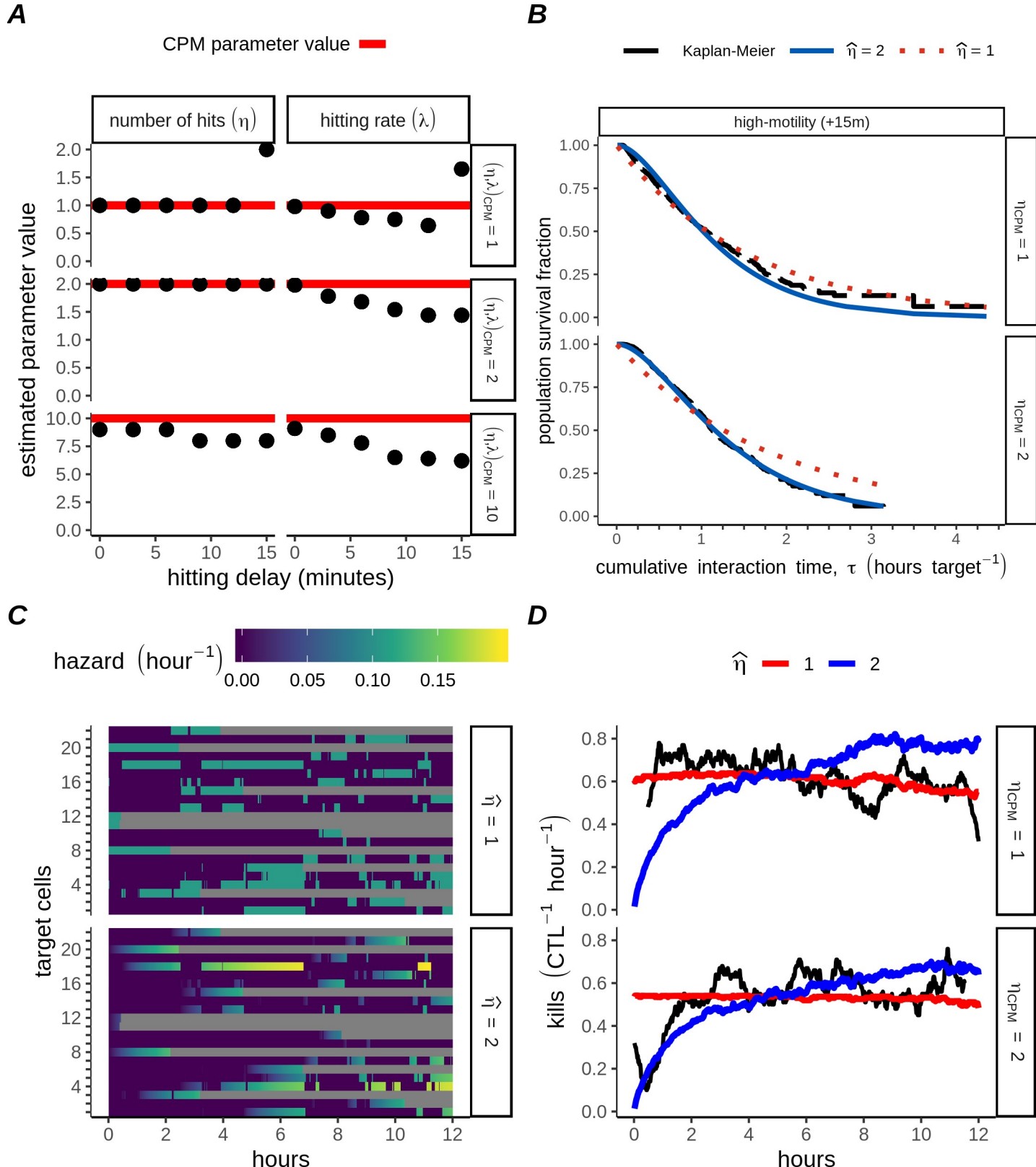

**Fig 6. Parameter retrieval for multiple-hitting CTLs with underlying hitting delay.** A) Estimated parameters (points), compared to the underlying parameter values used (red lines) in CPM simulations, featuring high-motility CTLs, in which we varied the lower bound for the time (in minutes, as indicated) needed for hitting. B)

Kaplan-Meier survival functions (black lines), or survival functions plotted with estimated parameters fitted to data from $N_w = 100$ CPM simulations generated by multiple-hitting (lower row, $\eta_{CPM} = 2$) or single-hitting CTLs (upper row, $\eta_{CPM} = 1$). C) Heatmaps from one CPM simulation containing high-motility, single-hitting CTLs who had a 15 minute lower bound set on the hitting time. Each row represents a single target. Target status is represented by colour: targets not-contacting the CTL are deep purple, and killed targets are grey. Contacted targets are coloured according to their momentary hazard according to two candidate parameter sets (top panel: $\hat{\eta} = 1, \hat{\lambda} = 0.65$; bottom panel: $\hat{\eta} = 2, \hat{\lambda} = 1.65$). D) Predicted killing rate according to two different candidate parameter sets ($\hat{\eta} = 1$, red lines; $\hat{\eta} = 2$, blue lines), candidates being themselves applied to CPM simulations with high-motility, +15m CTLs ($\eta_{CPM} = 1$, top panel; $\eta_{CPM} = 2$, bottom panel).

analysing the 'adjusted' cumulative contact durations between CTLs and individual targets, i.e. the total length of the interaction until either the target was killed or the experiment ended. Importantly, we found that measurements for both killed and surviving cells are required for this approach to be successful. This is because the limited time window of observation renders data that are in part censored, yet elapsed contacts that have not yet resulted in killed targets also contain information on underlying killing parameters. In a similar fashion, we previously developed a method to estimate absolute (i.e., not cumulative) cellular interaction times based on time lapse imaging data [23].

Although CTL cooperativity and multiple-hitting have now been described in a number of settings [7,10–12], a detailed quantitative description of the sequence of intracellular events which might underlie multiple-hitting does not yet exist. Several mechanisms can be envisaged which separately or collectively might result in target cells enduring sustained attacks from CTLs before death. A first factor which may explain the ability of target cells to endure sustained attacks is death occurring via the 'extrinsic apoptosis pathway', i.e., via tumour necrosis factor (TNF) or FAS-L. In the study of Vasconcelos et. al. [19], blockade of FAS-L did not diminish overall killing, suggesting that FAS-L was not involved in CTL killing. Moreover, separation of CTLs and targets in a transwell assay showed that contact was required before target cell apoptosis could occur. Although this result suggests that diffusible TNF did not contribute to target cell apoptosis, TNF is also expressed in transmembrane form [25] and may have contributed to contact-dependent killing, or could have synergised with other effector pathways. Furthermore, TNF or interferon-γ—another hallmark cytokine produced by activated CTLs— have been linked to an upregulation of FAS-L receptors in different cell types [26,27], or might otherwise synergise with FAS-L to induce target cell apoptosis [28]. The possibility of synergistic activators of the extrinsic apoptosis pathway is intriguing since activation of such mechanisms might explain delayed onset of burst-killing. This was observed in recent studies in which natural killer cells initially controlled tumour cell targets with a fast-acting, perforin-dependent mechanism, before switching to a mechanism primarily depending on engagement of death receptors [17,18]. It would be useful to investigate whether CTLs also utilise this mechanism.

A second factor which may account for multiple-hitting is heterogeneity in delivery of perforin and granzymes. Perforin alone induces rapid pore formation in target cell membranes, with such membrane disruption expected to increase the metabolic burden on target cells. Even if insufficient to directly induce apoptosis, one would expect such depletion to divert resources from adaptive cellular stress responses, thereby sensitising cells to death from other mechanisms. Granzymes are a diverse set of cytotoxic proteases with a broad array of intracellular targets [29]. A recent review highlights that perforin-mediated pore formation may or may not be accompanied by delivery of granzyme molecules into the cytosol [30]; a requirement for granzyme delivery appears to be the establishment of a sufficiently large pore at the point of contact between CTL and target. Examination of recent 4D images of CTL–target engagement highlight potential for heterogeneous delivery of cytotoxic molecules [8]. That study showed the capability of a single CTL to rapidly organise lytic molecules around the centrosome upon initial target recognition and to subsequently polarise the centrosome towards

the target. This sequence of events results in a strong and stable cytotoxic synapse with a high local density of perforin and granzymes. Anecdotal evidence from this same work indicates that there can also be an alternative outcome: In one observation a CTL attempted to form two immunological synapses with one target, with the result that effective centrosome polarisation towards either synapse did not occur and ultimately both synapses were aborted without target cell death (see S8 Video in reference [8]). Other observations of CTLs simultaneously polarising granules towards multiple targets [9] demonstrate that the formation of multiple immunological synapses does not necessarily preclude CTLs from killing. Taken together, these observations suggest that due to the diversity of possible damage pathways activated by CTLs as well as the potential for heterogeneity in delivery of granules, several mechanistic explanations for multiple-hit induced killing remain open.

Given the breadth of cytotoxic weaponry available to a single CTL, it is apparent that experimental interference with one or more CTL effector functions is insufficient to conclude that one or another pathway is primarily involved in target cell death in a given experiment. We suggest high resolution, *in vitro* imaging as an effective means of achieving insight into the CTL killing process. Such high-resolution imaging would have several benefits: clear visualization of the polarisation of the lytic granules towards target cells would allow acquisition of statistics regarding the lethality of hits. Moreover, monitoring of individual target cells over time would provide statistics regarding the formation and abortion rates of immunological synapses and regarding the probability of target cell death after multiple hits. In addition, such approaches would allow investigation of the possibility of target cell recovery between successive hits, along with assessing the timescale over which such recovery might occur. Although such spatio-temporal resolution might be challenging to achieve experimentally, recent approaches using structured environments [31,32] provide a possible means of achieving more refined control of CTL-target interactions.

In conclusion, in addition to recent efforts to further characterise heterogeneity amongst CTLs, greater attention is needed to simultaneous monitoring of mechanisms activated in target cells after the target has been contacted by a CTL, assisted by statistical analyses and computational methods such as those presented here. Experimental research particularly involving use of e.g. caspase-8 reporters or reporters of granzyme activity to compare the relative importance of different killing mechanisms, as recently done in NK cells [17,18] is crucial. Computational models can then be used to compare results between different experimental assays, thereby quantitatively assessing the contribution of identified CTL effector functions in different contexts.

## Methods

### Monte Carlo simulations

We devised stochastic simulations representing different "wells" in which individual CTLs killed targets. The setup of the simulations was based on published data by Vasconcelos *et al.* [19]. In brief, Vasconcelos *et al.* incubated pre-activated human-derived CTL clones with Epstein-Barr virus transformed B cell targets for 12 hours in microwells ($N_w$ = 259). Each microwell contained a single CTL confined with an indeterminate (approximately 10–20, see Fig 4A in reference [19]) number of targets. Microwells were approximately cylindrical and had a cross-section diameter of approximately 100μm. A caspase reporter was used to determine the killing rate of individual CTLs over time. Similarly, our simulations featured $N_w$ independent simulations, each containing $n$ initially unhit targets and lasting for a simulated time period of 12 hours, or until all targets had been killed. The simulations proceed as follows:

1. A random variable $x_{wait}$, representing the waiting time until the next CTL hit, is drawn from the exponential distribution with rate parameter equal to the CTL hitting rate $\lambda$. The current simulation time is increased by $x_{wait}$.

2. A random target is selected and its number of hits is increased by one.

3. If a target has received sufficient hits for death (i.e., $\eta$ hits), it is immediately removed from the simulation.

In some simulations, we extended the rules in order to reflect typical *in vitro* assays more accurately:

1. ***Variable target numbers.*** Each simulation contained a single CTL and a variable number of targets $n$. For each well the number of targets was drawn from a Poisson distribution with mean $\bar{n}$.

2. ***Variable hitting rate.*** For each simulation the hitting rate $\lambda$ of each CTL was a normally distributed random variable. The standard deviation of this distribution was used as a model parameter, with larger standard deviation reflecting CTL populations with greater intrinsic heterogeneity in killing performance between individuals.

3. ***Dynamic conjugate formation.*** We considered that hit delivery had to be preceded by conjugate formation and that at $t = 0$ *hrs* the CTL has not yet encountered any targets, and that CTLs form new conjugates with targets at constant rate $k_{on}$ and abort conjugates with constant rate $k_{off}$. Thus these simulations consider 4 distinct types of event: in addition to hitting and dying, we now have conjugate formation and conjugate abortation. The Gillespie algorithm was used to determine the type of event and waiting time between subsequent events [33], except for target cell death which occurs immediately after the lethal hit just as in our "all at risk" simulations.

Parameter estimation for the dynamic conjugate formation model was based on four reported values from Vasconcelos *et. al.* [19]: the population killing averages for the high rate killers (6.4 targets killed per 12 hours) or low rate killers (2.8 targets killed per 12 hours), the fraction of the population reported to be high rate killers (⅓), and the breakpoint after which the high rate phenotype appeared (8–10 hours; we took 9 hours for this value). From these 4 reported values we derived three statistics for fitting our Monte Carlo simulations with dynamic conjugate formation: the mean ($a_1$) and variance ($a_2$) of the number of killed targets per CTL after 12 hours, and the expected number of killed targets per CTL at the breakpoint of 9 hours ($a_3$). We estimated the killing at 9 hours by noting that the high rate group had not yet emerged at 9 hours, before which all cells killed at an approximately constant rate. Thus extrapolating from the low rate killing average at 12 hours (2.8 x 9/12) gives approximately 2 targets killed at the 9-hour breakpoint (note that this is also consistent with Fig 4B of Vasconcelos et. al [19]). Thus, the experimental estimates were: $a_1 = 4$, $a_2 = 6.9$ and $a_3 = 2$. To fit to these estimates, we measured the same statistics ($b_{1,2,3}$) from our simulations and then minimised the root mean squared error:

$$RMSE = \sqrt{(1/3 \cdot \sum_{i:1,2,3}(a_i - b_i)^2)}, \qquad\qquad \text{Eq 1}$$

for different values of the parameters $\eta$, $\lambda$, $k_{on}$, and $k_{off}$. For the stochastic optimisation we performed 10 repeats for all combinations of selected discrete values of $\eta$, $k_{on}$, and $k_{off}$ (S2 Fig), and then for each combination we estimated $\lambda$ based on $N_w = 1000$ repeats and the optimise function in R. Dynamic conjugate simulations were written in C++ using the Rcpp package.

**Table 1. Stochastic simulation parameters.**

| parameter | biological interpretation |
|---|---|
| $\eta$ | number of hits required for target death |
| $\bar{n}$ | mean number of targets in a well |
| $\lambda$ (hr$^{-1}$) | hitting rate |
| $k_{on}$ (hr$^{-1}$) | conjugate formation rate |
| $k_{off}$ (hr$^{-1}$) | conjugate dissociation rate |

Biological interpretation of parameters for the stochastic simulations are summarised in Table 1 and the parameter values used throughout the manuscript are provided in S1 Table.

## Spatial simulations

We developed spatial simulations of CTLs killing in microwells, with the aim of generating noisy and undersampled artificial data representative of data generated by microscopy, data which can be used to test methods for recovery of parameters governing CTL hitting. To this end we employed the cellular Potts model (CPM) framework [34], a formalism we used previously to simulate T cell-target cell interactions [11,35,36]. The CPM is a lattice based model, with entities such as cells represented by assigning individual lattice sites a 'spin' value, to identify them as belonging to a specific entity. The model evolves via minimisation of an energy function, the Hamiltonian:

$$H = H_{sort} + H_l + H_{act}. \qquad \text{Eq 2}$$

Here, $H_{sort}$ represents interactions between cell surfaces and deviations from a target cell area; $H_{sort}$ is defined as [34]:

$$H_{sort} = \sum_{(a(\sigma)-A_{q(\sigma)})^2} J\big(q(\sigma(i,j)), q(\sigma(i',j'))\big)\big(1 - \delta_{\sigma(i,j),\sigma(i',j')}\big) + \zeta_a \sum_{\text{spin types } \sigma} \big(a(\sigma) - A_{q(\sigma)}\big)^2, \quad \text{Eq 3}$$

where $\sigma(i,j)$ is the spin of an individual cell of type $q$ at grid point with $x$ coordinate $i$ and $y$ coordinate $j$; $J(q,q')$ is the surface energy between cells of type $q$ and $q'$; $\delta_{\sigma,\sigma'}$ represents the Kronecker delta; $a(\sigma)$ represents the actual area of a cell and $A_{q(\sigma)}$ the target area for a cell of type $q$ (we refer to this as area rather than volume because we employ 2D simulations); $\varsigma_a$ is a weighting term for the area constraint; Note that the sum of the surface energies are calculated over each third order neighbour of a 2D grid site.

Our model also includes a term for surface area conservation of individual cells [37]:

$$H_l = \zeta_l \sum_{\sigma} \big(l(\sigma) - L_{q(\sigma)}\big)^2, \qquad \text{Eq 4}$$

where $L_{q(\sigma)}$ is the target perimeter for cells of type $q$, $l(\sigma)$ is the current perimeter of a cell with type $\sigma$ (determined as the total length of the boundary interfaces with grid sites of differing spin), and $\zeta_l$ is the weight of the perimeter constraint. We set $L_q = 2\pi\sqrt{A_q}$, i.e., the ratio of a circle's perimeter to its area, so that the term $H_l$ is minimised when cells become perfectly circular. We set $\zeta_l$ lower for the CTLs than for the target cells, implying that the targets retained a spherical shape whereas CTLs were much more deformable in our simulations.

Finally, the Hamiltonian includes a term $H_{Act}$ to drive the motility of CTLs [22]:

$$H_{Act} = \frac{\varsigma_{Act}}{Max_{Act}}\big(GM_{Act}(u) - GM_{Act}(v)\big). \qquad \text{Eq 5}$$

This follows an actin-driven cell motility model with protrusions driving the migration of cells. In this model actin is modelled explicitly and when a cell occupies a new site on the lattice, the site is given an actin value $Max_{Act}$. The actin activity $Act$ in that site then decreases by one at every Monte carlo step until it reaches 0. The function:

$$GM_{Act}(u) = \left(\prod_{y \in V(u)} Act(y)\right)^{1/|V(u)|} \qquad\qquad \text{Eq 6}$$

calculates the geometric mean actin activity around site $u$, where $|V(u)|$ are the second order Moore neighbours of site $u$ (see Fig 1 of reference [22]). The model favours updates from sites $u$ with high actin activity into neighbouring sites $v$ with low actin activity, resulting in local positive feedback. The CPM parameter $\varsigma_{Act}$ is a weighting term the strength of which we varied to control the motility of the CTLs. The $H_{act}$ term was not applied to target cells, which are moved only passively via interactions with the CTL and other targets.

In our spatial simulations we also implemented a contact-limited hitting behaviour for the CTL. We take CTL killing of targets to occur primarily via the perforin/granzyme pathway so we consider only contacted targets to be at risk, although our model should also apply to FAS-Ligand mediated killing, which is also contact-limited. When multiple targets are contacted by a CTL, it seems likely that the risk of getting hit is not equal for all targets, as polarisation of the lysosome towards specific targets should occur in order to permit delivery of lytic molecules to the target [8,9]. Although we did not model the polarisation of the lysosome explicitly, we do take into account a tendency for CTLs to unequally distribute hits towards contacted targets. To achieve this, we implement the same baseline hitting probability as in the Gillespie simulations, and multiply this by $\theta_i(t)$, the proportional fraction of CTL: target membrane interface occupied by the target at time point $t$:

$$\theta_i(t) = \frac{l_i(t)}{L_i(t)}, \qquad\qquad \text{Eq 7}$$

where $l_i(t)$ is the length of the interaction interface between target $i$ and the CTL inhabiting the same well, $L_i(t)$ the total interaction interface length of the CTL that contacts target $i$, including any other co-contacting targets. Because CTLs are considered to hit targets at a constant rate $\lambda$, for simulations without delayed hitting each target's risk of being hit during a brief time interval $\Delta t$ equals $\lambda\theta(t)\cdot\Delta t$. For some simulations we introduced a rule preventing CTLs from hitting targets for a specified delay period each time a CTL contacted or recontacted a target. This was implemented by means of a counting variable inside each target, such that hits would not register until the target had been in continuous contact with the CTL for the specified interval.

Simulations had a spatial scale of 1 μm pixel$^{-1}$ and were 100 μm$^2$ in area. The simulation space consisted of a circular area representing a microwell within which one CTL and usually between 10–20 targets were constrained to move. Simulations had a temporal scale of 1 second per Monte Carlo step. Parameters employed in the CPM simulations are given in Table 2.

**Table 2. Cellular Potts simulation parameters.**

| parameter | value | Description |
|---|---|---|
| $J_{\sigma,\sigma'}$ | $J_{tar,tar} = 0.7$; $J_{ctl,tar} = -3$; $J_{tar,well} = 0$; $J_{ctl,well} = 0$ | surface energies between cell types |
| $A_q$ | $A_{ctl} = 140 \ \mu m^2$ $A_{tar} = 340 \ \mu m^2$ | the target area for a cell of type $q$ |
| $L_q$ | $2\sqrt{\pi A_q}$ | the target perimeter for a cell of type $q$ |
| $\varsigma_l$ | $\varsigma_{l,ctl} = 0.1$ $\varsigma_{l,tar} = 0.25$ | strength of cell perimeter constraint |

(*Continued*)

**Table 2.**  (Continued)

| parameter | value | Description |
|---|---|---|
| $\varsigma_a$ | $\varsigma_{a,ctl} = 1$ <br> $\varsigma_{a,tar} = 1$ | strength of cell area constraint |
| $\varsigma_{Act}$ | $\varsigma_{Act,low} = 2$ <br> $\varsigma_{Act,high} = 10$ | strength of actin protrusion dynamics: $\varsigma_{Act,low}$ for low-motility and $\varsigma_{Act,high}$ for high motility CTLs |
| $Max_{Act}$ | 50 | Actin activity value when CTLs occupy a new lattice site |

Simulation output was produced every 120 Monte Carlo steps (2 minute intervals), corresponding to a typical sampling frequency in time-lapse imaging data with multiple wells [19]. CPM simulations were developed within the morpheus framework [38].

## Supporting information

**S1 Table. Summary of parameters used in stochastic simulations.** Data and code used in this project are available (http://doi.org/10.17605/OSF.IO/6GQYP).
(PDF)

**S1 Text. Fitting procedure and hypothesis comparison for multiple-hitting model and subpopulation model.**
(PDF)

**S1 Fig. Multiple hitting increases inherent variability in killing performance between individual CTLs.** A-B) Distribution of killed target numbers after 12 hours (A) when intrinsic hitting rates $\lambda$ (B) are drawn from a normal distribution with mean $\bar{\lambda}$ and standard deviation $\sigma_\lambda (\lambda \sim Normal(\bar{\lambda}, \sigma_\lambda))$. C) Overdispersion for the variance in killed targets in A relative to the variance expected for a Poisson distribution, i.e., the ratio of the variance ($var(x)$) to the mean ($\bar{x}$) number of targets killed after 12 hours (vertical axis). The horizontal axis is the ratio of the standard deviation to the mean value of the intrinsic hitting rate.
(TIFF)

**S2 Fig. Parameter estimation for Monte Carlo simulations with dynamic contacts.** A) Estimated hitting rates ($\lambda$, represented by colour) for various combinations of the number of hits ($\eta$, rows), contact formation rates ($k_{on}$, vertical axes in sub-panels), or contact escape rates ($k_{off}$, horizontal axes in sub-panels). Ten repeats (across columns) were performed for the optimisation step, using $N_w = 10^3$ CTLs per tested value of $\lambda$. After fitting we validated our results by performing $N_w = 10^4$ simulations with each best fitting parameter combination, which is shown here. B) Root mean square residual errors for the best fitting parameter estimates (panel arrangement is as described in S2A Fig legend). Results are from validation simulations, using $N_w = 10^4$ simulations per parameter combination.
(TIFF)

**S3 Fig. Monte Carlo simulated CTL:target interaction durations amongst surviving and killed targets.** A) Sample density of killed targets in Monte Carlo simulations lasting until all targets were killed, with different numbers of hits ($\eta$, on different rows). B) Sample density of killed targets in Monte Carlo simulations stopped after 12 hours. C) Sample density of surviving targets, corresponding to the 'absent' portion of the distribution for killed targets in B. The red line in A and B is the function $f_k$, which describes how the relative probability until targets receive η hits arriving at a constant rate λ depends on the cumulative interaction time τ, for a gamma distributed waiting time. For all S3 Fig: $N_w = 100$, n = 12 targets per well, all targets

equally at risk. Parameter combinations used were: ($\eta = 1, \lambda = 0.34$; $\eta = 2, \lambda = 1.17$; $\eta = 3, \lambda = 2.12$; $\eta = 4, \lambda = 3.14$; $\eta = 5, \lambda = 4.22$).
(TIFF)

**S4 Fig. Maximum likelihood estimation for the killing rate of single-hitting CTLs.** A) Poisson distributions for the number of targets used to start simulations in S4 Fig, with mean $\bar{n} = 8$ *or* 16 as shown. B) Number of killed targets after 12 hours for $N_w = 2 \times 3 \times 1000$ simulations, each group of $N_w = 1000$ started with one of the 2 distributions in A, and with one of the 3 indicated parameter settings. C) Density of killed targets after 12 hours from 'Mixed' distributions resulting from $\eta = 1, \lambda_{LR} = \mathbf{0}.2, \lambda_{HR} = 0.7$ and either $\bar{n} = 8$ *and* $m = 0$ (left panel), or $\bar{n} = 16$ *and* $m = 0.67$ (right panel). Note that for $\bar{n} = 8$ the killing of multiple-hitting CTLs became greater than the high rate subpopulation of single-hitting CTLs; $\bar{n} = 8$ was only used for testing robustness of the estimators on heavily censored data. D) Relative likelihood of candidate hitting rate estimates, $\hat{\lambda}$, compared to the maximum likelihood estimate, $\hat{\lambda}_{ML}$, resulting from application of the Poisson estimator separately to each of the single-hitting ($\eta = 1$) datasets shown in B. Relative likelihood are shown either for the dataset in its entirety (dashed lines), or for a randomly selected sample of $N_W = 10$ (solid lines). E) Examples of testing datasets derived from the multiple-hitting population (B, $\bar{n} = 16$, $\eta = 10$) or from a mixture of single-hitting CTLs (B, $\bar{n} = 16$, $\eta = 1$, where the true density of killed targets in the mixture distribution is in C). F) Relative likelihood of candidate hitting rate estimates, $\hat{\lambda}$, compared to the maximum likelihood estimate, $\hat{\lambda}_{ML}$, for constrained fits constructed from either the subpopulation datasets, or from multiple-hitting datasets, for three samples with either $N_w = 30, 100$, or 1000 (note the multiple-hitting-generated data ($\eta = 10$) is therefore fully represented by the $N_w = 1000$ case).
(TIFF)

**S5 Fig. Testing for multiple-hitting CTLs versus subpopulations of single-hitting CTLs.** A) Maximum likelihood estimates for the hitting rate, $\hat{\lambda}_{ML}$, with either the gamma or Poisson estimators, both constrained to a uniform single-hitting population (i.e. by forcing $\eta = 1$ for the gamma estimator and by forcing $m = 1$ for the Poisson estimator). Each of the $2 \times 4 \times 10 = 80$ points represents one of the $4 \times 10$ testing populations from S4E Fig (here indicated by facet labels), fit with both of our estimators (x-axis). B) Difference between the log likelihood function evaluated with the constrained versus unconstrained Gamma estimator $\log L(\hat{\lambda}_{ML}, \hat{\eta}_{ML})$ (dark bars); or with the constrained versus unconstrained Poisson estimator $\log L(\hat{\lambda}_{HR,ML}, \hat{\lambda}_{LR,ML}, \hat{m}_{ML})$ (light bars). Each of the 40 testing populations occupies one horizontal bar, with the details of the testing populations as indicated in facet labels. For the x-axis scaling (negative values are not possible), the relative size of the dark v.s. light bars is proportional to the strength of the evidence for the multiple-hitting hypothesis (dark bars) versus the subpopulation hypothesis (light bars). C) The constrained estimates for the hitting rate parameters, $\hat{\lambda}_{ML}$, (circles; also shown in A) or their unconstrained counterparts (red asterisks) for each testing population (points on x-axis). For the Gamma estimator (top row) the estimated $\hat{\eta}_{ML}$ is shown only where $\hat{\eta}_{ML} > 1$. For the Poisson estimator (bottom row), the unconstrained estimates for $\hat{\lambda}_{HR,ML}, \hat{\lambda}_{LR,ML}$ are above and below their counterpart constrained estimates, and the Gamma and Poisson estimators can be compared per population. D) Distribution of all cumulative interaction times, $\tau$ (killed and surviving targets shown separately in columns), for all $N_w = 1000$ members of each of the 3 generating populations (as shown in S4B Fig). Although the 2 single-hitting populations were combined (upper row), the separate contribution of the $\lambda_{HR}$ (red) or $\lambda_{LR}$ (blue) populations is indicated by

color. Multiple-hitting CTLs (green) are shown separately (bottom row).
(TIFF)

**S6 Fig. Maximum likelihood estimation for CPM simulations with a hitting threshold time.** A-B) Heatmaps of the likelihood function around the maximum likelihood estimates for the killing parameters $\eta$ and $\lambda$, in CPM simulations under various conditions. In A, results are shown for various $\eta$ values (rows) and for both high-motility (left colum) and low-motility (right column) conditions for simulations with 15 minute hitting delay. In B, results are shown for high motility CTLs at all tested values of the delay (in range 0–15 minutes, across columns). The horizontal and vertical lines in A-B mark the values of the CPM parameters used to generate the data for each group and the boundary enclosing the 95% confidence region is marked with a thin blue line.
(TIFF)

**S1 Video. Simulation of high-motility CTL, requiring 5 hits to kill targets.**
(AVI)

**S2 Video. Simulation of low-motility CTL, requiring 5 hits to kill targets.** In all videos, CTLs are shown in red whilst uncontacted targets are in grey. Contacted targets are shaded blue based on their share of total CTL:target interface, i.e. the probability that they will receive the next hit generated by the CTL. Targets are overlaid with the number of hits they have received. Elapsed simulation time (hours:minutes) is displayed in the upper right corner of the videos.
(AVI)

## Acknowledgments

We would like to thank Delphine Guipouy and Loïc Dupré for useful discussions on the set-up of previously published microwell experiments.

## Author Contributions

**Conceptualization:** Richard J. Beck, Dario I. Bijker, Joost B. Beltman.

**Formal analysis:** Richard J. Beck.

**Funding acquisition:** Joost B. Beltman.

**Investigation:** Richard J. Beck.

**Methodology:** Richard J. Beck, Dario I. Bijker.

**Project administration:** Joost B. Beltman.

**Software:** Richard J. Beck.

**Supervision:** Joost B. Beltman.

**Validation:** Richard J. Beck.

**Visualization:** Richard J. Beck.

**Writing – original draft:** Richard J. Beck.

**Writing – review & editing:** Joost B. Beltman.

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
