## [Decision Letter · Decision Letter 0]

12 Feb 2020

Dear Dr. Beltman,

Thank you very much for submitting your manuscript "Heterogeneous, Delayed-Onset Killing by Multiple-Hitting T Cells: Stochastic Simulations to Assess Methods for Analysis of Imaging data" for consideration at PLOS Computational Biology.

As with all papers reviewed by the journal, your manuscript was reviewed by members of the editorial board and by several independent reviewers. In light of the reviews (below this email), we would like to invite the resubmission of a significantly-revised version that takes into account the reviewers' comments.

We cannot make any decision about publication until we have seen the revised manuscript and your response to the reviewers' comments. Your revised manuscript is also likely to be sent to reviewers for further evaluation.

Sincerely,

Andrew J. Yates

Associate Editor

PLOS Computational Biology

Rob De Boer

Deputy Editor

PLOS Computational Biology

Reviewer's Responses to Questions

**Comments to the Authors:**

Reviewer #1: Comments:

Beck et al simulate CTL-mediated immune responses in different settings to better understand how CTL-mediated killing of target cells is regulated in vitro.

They use different simulation approaches to address the very important open question of how CTL-mediated killing of target cells can be better understood. This remains a very important question, given the high clinical importance of CTL-mediated immune responses against infectious diseases and cancer in the current age of immune-cell based therapeutic approaches.

Reviewer comments regarding the text:

Regarding the Monte Carlo Simulations in Figure 1, it remains a little unclear how a small increase in simulated target cell numbers (plus 1) would change the outcome of the simulation in the way presented in Figure 1B (for the 100 hit case). Obviously, the “100 hit case” is a very extreme situation – but maybe this outcome of the simulation could be discussed and explained in more detail.

Maybe a plotting of the effector-to-target cell ratio of these simulations would help to better understand this result?

Regarding the “over-dispersion” – maybe a short explanation of how such a conclusion was reached might be helpful (how was dispersion measured, and how can the cut-off that determines over-dispersion be justified).

Regarding the spatial model, it remains unclear how the “two CTL motility patterns” are used in the simulations: how are the two states set for individual cells? Does target cell contact decrease motility? Based on what experimental data are these motility features defined?

Finally, to better understand how “cumulative contact time” in the simulations can be interpreted a short description of how the simulated CTL deliver the “lethal hits” in relation to contact duration would be helpful. How is “hit delivery” defined in the spatial model? Could one contact event lead to multiple “hits”?

Reviewer comments on Material and Methods section:

1.) Monte Carlo simulations:

How were these simulations performed? How was the data from Vasconcelos et al used exactly and where can the used data be found?

How many simulations were performed?

2.) Spatial simulations

How was target cell motility defined? Why are the targets motile – how much does that affect the different conclusions (Maybe check results with non-motile and motile targets).

Reviewer comments on Figures:

Figure 1:

Fig 1B: what does the Y-axis label really mean (“frequency”)? Maybe this can be described in the figure legend.

Fig 1D: The Y axis is cut at 10 – is this really necessary?

Fig 1E: what does the Y-axis label mean – is it different to Fig 1B?

Figure 2:

Usually, target cell binding is expected to alter CTL migration – so how does target binding affect the CTL motility here?

How are the high vs low motility states “regulated” in these simulations?

Reviewer #2: In this study, the authors use stochastic simulations to examine if heterogeneity in CTL killing dynamics can be explained by the requirement of target cells to receive multiple lytic hits. Using explicit spatial simulations based on the CPM-framework, they further investigate how this hitting rate and the number of required hits per target cell can be inferred from imaging data. Based on their analyses they show that measuring the cumulative duration of CTL-target contacts would improve the accuracy of estimating CTL killing kinetics.

The topic is timely and relevant as understanding CTL efficacy is an important aspect for designing optimal therapeutic vaccination and treatment strategies, as e.g. against cancer. The study is well written and thoroughly structured. With this theoretical analysis the authors provide a reasonable explanation for previous observations from experimental studies and propose required measurements for further experiments. However, the study remains a bit limited by focusing on only one explanation for CTL killing dynamics.

# Major points:

(1.) The authors state that with the multiple-hitting-scenario, there is no need to invoke an unobserved subpopulation of high-rate killers (p. 22) as suggested by other studies. While this is true and nicely shown, I think the study would substantially benefit if the authors also show how the proposed information from imaging studies should differ in case of the latter scenario or others. It could be quite interesting to know on what type of measurements one has to focus in order to differ between different hypotheses, such as having cell populations that are heterogeneous (either in their killing efficacy (CTL) or susceptibility to killing (target cell)), having serial or multiple-killing, or assuming single- or multiple hit killing.

(2.) The ability to infer the hitting parameters in case of multiple-hitting CTL from CTL-target (cumulative) contact times in case of multiple-hitting is quite robust and interesting. However, measuring the cumulative contact times might be not sufficient in case that active conjugate formation would require the stopping of CTLs to allow them to deliver their lethal hits (Wiedemann et al., PNAS 2006), i.e., requiring a minimal contact time. This seems not to be directly implemented within the CPM as the hitting probability only considers the contact length for distributing the hit across multiple target cells.

(3.) Another question concerns the parameterizations of the CPM. Do “high-“ and “low-motility” CTL correspond in some way to the dynamics observed within the experiments or were they arbitrarily chosen?

(4.) Why is the average number of target cells killed after 12 hours for a density of 15 target cells higher when several hits are required compared to only one? This seems to be counterintuitive unless the hitting rates differ between the scenarios, maybe due to the adjustment done before and mentioned in the text? However, Figure 1E actually shows that for a given target cell concentration, the distribution gets shifted to the left the more hits are required. The values of the rates used for simulation should be mentioned within the text or the figure legends. The corresponding Table (Table 1) seems to be empty.

# Minor points:

- Figure 4A and Figure 5F are very difficult to read. Maybe there are better ways of representation, e.g. using different colors for each given (correct) parameter combination and similar colors with point sizes corresponding to frequencies for the estimates?

- Figure 1B: Does the red dotted line always refer to the 16 targets case for each row separately, i.e. for each number of hits (1,10,100)?

**Have all data underlying the figures and results presented in the manuscript been provided?**

Reviewer #1: None

Reviewer #2: No: The parameters used for simulating the results shown in Figure 1 seem to be missing.

PLOS authors have the option to publish the peer review history of their article (what does this mean?). If published, this will include your full peer review and any attached files.

Reviewer #1: No

Reviewer #2: No
---

## [Editor Report · Decision Letter 1]

21 May 2020

Dear Dr. Beltman,

We are pleased to inform you that your manuscript 'Heterogeneous, Delayed-Onset Killing by Multiple-Hitting T Cells: Stochastic Simulations to Assess Methods for Analysis of Imaging data' has been provisionally accepted for publication in PLOS Computational Biology.

Best regards,

Andrew J. Yates

Associate Editor

PLOS Computational Biology

Rob De Boer

Deputy Editor

PLOS Computational Biology

---

## [Editor Report · Acceptance letter]

22 Jun 2020

PCOMPBIOL-D-19-02233R1 

Heterogeneous, Delayed-Onset Killing by Multiple-Hitting T Cells: Stochastic Simulations to Assess Methods for Analysis of Imaging data

Dear Dr Beltman,

I am pleased to inform you that your manuscript has been formally accepted for publication in PLOS Computational Biology. Your manuscript is now with our production department and you will be notified of the publication date in due course.

With kind regards,

Laura Mallard
